# Psychological Factors Influencing Achievement of Senior High School Students

**DOI:** 10.3390/healthcare10071163

**Published:** 2022-06-22

**Authors:** Nongluck Kienngam, Narong Maneeton, Benchalak Maneeton, Pichaya Pojanapotha, Jutipat Manomaivibul, Suttipong Kawilapat, Suntonrapot Damrongpanit

**Affiliations:** 1Division of Educational Psychology and Guidance, Department of Educational Foundations and Development, Faculty of Education, Chiang Mai University, Chiang Mai 50200, Thailand; nongluck.kienngam@cmu.ac.th; 2Department of Psychiatry, Faculty of Medicine, Chiang Mai University, 110 Intawaroros Road, Sriphum, Muang, Chiang Mai 50200, Thailand; benchalak.maneeton@cmu.ac.th (B.M.); pichaya.poj@cmu.ac.th (P.P.); jutipat.m@gmail.com (J.M.); suttipong.kawilapat@cmu.ac.th (S.K.); 3Division of Educational Evaluation and Research, Department of Educational Foundations and Development, Faculty of Education, Chiang Mai University, Chiang Mai 50200, Thailand; suntonrapot.d@cmu.ac.th

**Keywords:** depression, anxiety, self-esteem, life satisfaction, academic achievement, interpersonal relationships

## Abstract

Numerous factors are proposed to affect high school students’ academic achievement; however, these factors may not reveal all possible causal relationships. This study conducted path analysis to examine the direct and indirect effects of interpersonal relationships, life satisfaction, self-esteem, anxiety, and depression on the academic achievement of senior high school students. Two hundred and eighty-five students from five schools in Chiang Mai, Thailand, aged 14–19 years, were included for data analysis. The fit indices of all models were in agreement with the empirical data. Anxiety levels had a significantly positive direct effect on achievement, whereas depression had a negative direct effect on achievement. Additionally, self-esteem, life satisfaction, and interpersonal relationships had negative indirect effects on depression and anxiety. A program that stimulates the optimal and appropriate level of anxiety may be useful. An appropriate level of anxiety appeared positively related to academic achievement, but a high level of anxiety relatively influenced the incidence of depression. Thus, encouraging self-esteem, interpersonal relationships, and life satisfaction can promote academic ability and decrease the risk of depression. Further well-designed and large sample-size studies should be conducted to confirm these findings. The interplay of all studied factors may account for the variation in academic achievement, depression, and anxiety of 11.60%, 42.80%, and 17.60%, respectively.

## 1. Introduction

Academic achievement usually relies on individual intellectual and cognitive ability. However, many students encounter problems with learning performance. In addition to the efficacy of educational programs, students’ non-academic attributes, including physical and mental health, can also influence learning performance [1]. Depression, which is highly prevalent among students, can decrease academic achievement [2]. To our knowledge, the causal relationship between achievement and depression is displayed in different dimensions [3,4,5,6]. Anxiety plays an essential role in both academic achievement [3,6,7,8,9,10,11] and depression [2,9,12,13,14]. In addition to depression and anxiety, previous studies revealed that several psychological factors are also associated with academic achievement, including interpersonal relationships [15,16,17,18,19] and self-esteem [19,20,21].

Previous studies also illustrate that interpersonal relationships [13,22,23,24,25], self-esteem [13,14], and life satisfaction [26,27] are significantly associated with depression and anxiety. Moreover, the intercorrelation of these factors was proposed in several studies. Interpersonal relationships were associated with self-esteem [13,14,19,28] and life satisfaction [29,30]. The influence of life satisfaction on anxiety [26] and self-esteem [27] were presented in a previous study. Additionally, self-esteem was also associated with anxiety [14].

The overlapping associations of self-esteem, life satisfaction, and interpersonal relationships with depression and anxiety may be mediators contributing to their influences on academic achievement. Previous studies which focused on finding factors that directly influence academic achievement may lead to the neglect of some indirect associations. Therefore, the aim of this study is to illustrate the association of study achievement with the potential influence of psychological factors among high school students using path analysis, which facilitates investigation of both the direct and indirect effects of each factor.

### Literature Review

Using previous studies, we researched the association of study variables with academic achievement, as well as study variables with each other, in different dimensions. We then conducted a literature review of the association of studied factors with achievement, and each other, to build the hypothetical model used in this study.

Depression is a mood disorder that causes a persistent feeling of sadness and loss of interest. It often presents with physical symptoms, primarily fatigue, pain, or sleep disturbance. A depressed mood may or may not be present. Depression can play an essential role in affecting academic achievement. A previous study among secondary school students in Iran found that students with a high level of depression had a low academic achievement [3]. Several studies also show that depression and academic performance correlate significantly [4,5,6,31]. Interestingly, depression was also shown to have a positive relationship with academic performance among students with low social support in a previous study [5].

Anxiety can have both positive and negative effects on academic achievement. An appropriate level of anxiety positively enhances work efficiency, including learning ability, while a high level of anxiety may negatively affect academic achievement [9,22]. A previous study found that students with low levels of academic achievement tend to have higher levels of anxiety, depression, and stress [6,12].

Adolescents’ interpersonal relationships are one interesting exogenous variable related to mental health and academic achievement. Previous studies illustrated that adolescents with interpersonal relationship difficulties have a high risk of depression. Meanwhile, positive childhood experiences such as being part of a school, close relationships with peers, and good family support can prevent adolescents from developing depression and anxiety [13,22,23]. Additionally, good relationships among family members increases academic achievement in adolescents [19]. Previous studies also suggested that peer relationships are significantly related to academic achievement [15,16,17,18].

Life satisfaction also plays a role related to depression. A study of college students found that life satisfaction was negatively associated with depression [26]. Interestingly, self-esteem was positively correlated with life satisfaction [27].

Generally, self-esteem is the attitude of a person who is satisfied or dissatisfied with themselves. Maslow classified esteem needs into two categories: (i) esteem for oneself, which is an internal quality comprising dignity, achievement, mastery, and independence; and (ii) the desire for a reputation or respect from others. Maslow indicated that the need for respect or reputation is most important for children and adolescents and that it precedes genuine self-esteem or dignity [32]. Self-esteem was also negatively correlated with the incidence of depression, anxiety, and suicidal ideation [13,14]. Adolescents with high self-esteem are less likely to experience depression than those with low self-esteem. A previous study of high school students found that educational stress and domestic violence are risk factors for low self-esteem [14]. Additionally, academic achievement and self-esteem were highly positively correlated [21]. Additional studies illustrate a significant positive correlation between self-esteem and academic achievement. High self-esteem increases students’ academic achievements [19,20].

## 2. Materials and Methods

### 2.1. Hypotheses

According to the synthesis of documents and related research, the factors influencing academic achievement among high school students considered for inclusion in this study are depression, anxiety, self-esteem, life satisfaction, and interpersonal relationships. We hypothesized that the negative influence factors for academic achievement are depression (H1) and anxiety (H2), and that the positive influence factors for academic achievement are interpersonal relationships (H3) and self-esteem (H4). Additionally, we hypothesized that the positive influence factors for anxiety (H5) and the negative influence factors for depression include interpersonal relationships (H6), life satisfaction (H7), and self-esteem (H8). We further hypothesized that the negative influence factors for anxiety include interpersonal relationships (H9), life satisfaction (H10), and self-esteem (H11). As previous studies illustrated intercorrelation among these factors, the authors additionally hypothesized the positive influence of interpersonal relationships (H12) and life satisfaction (H13) on self-esteem, and the positive influence of interpersonal relationships (H14) on life satisfaction (Table 1). We aimed to examine both the direct effects in all 14 hypothesized pathways as well as all plausible indirect effects of each factor and other mediators.

### 2.2. Participants and Settings

The participants were senior high school students in the 2021 academic year who were randomly selected from five schools in Muang district, Chiang Mai, Thailand, based on the type of school (private, government, or demonstration high school). Following this, samples from each school were selected using stratified random sampling by study year and sex. The sample size was obtained using the formula for proportion estimation as follows:$$n=\frac{n_{0}}{1+\frac{n_{0}}{N}}$$
where *n* = required sample size; *n_h_* = required sample size from each grade = *nW_h_*; *N* = total number of senior high school students in Muang District, Chiang Mai Province (11,448); *N_h_* = total participants in each grade (*N*_1_ = 1805, *N*_2_ = 1804, *N*_3_ = 1760); *n*_0_ = $\frac{1}{V}\overset{L}{\underset{h=1}{\sum }}W_{h}p_{h}q_{h}$; *W_h_* = weight for each grade = *N_h_*/*N*; *h* = identification number of grade (1, 2, and 3 for grades 10, 11, and 12, respectively); *p_h_* = estimated prevalence of depression (21%) according to previous study [33]; *q_h_* = 1 − *p*; *q* = 1 − 0.21 = 0.79; *V* = variance of proportion estimation according to acceptable sample error with 95% confidence interval = (*d*/*Z*)^2^; *Z* = the standard normal deviation corresponding to a significance criterion of 0.05 = 1.96; and *d* = acceptable sample error which is assumed to be 0.05.

However, to account for sampling error, we increased the sample size to 10%; the total required participants were 275 (94 from grade 10, 91 from grade 11, and 90 from grade 12).

In addition, the number of parameters of the present hypothesis model was also determined for the required sample size. Since this model sets a total of 24 values parameters for the path analysis, the minimum sample size required (5–10 sample numbers per estimated parameter [34]) should be 120–240 subjects.

### 2.3. Data Collection and Assessment

#### 2.3.1. Demographic Data

We collected demographic data such as sex, age, study year, type of high school program curriculum, careers of family members, type of residence, history of mental illness, alcohol and substance use, and socioeconomic status. We also gathered additional information including daily life events and relationships, weekly study duration (for both regular and extraordinary courses), plans for future study and career, average grade points, study satisfaction, school burdens, stress management, and relationships with peers, family members, and teachers. The data were collected using a questionnaire from August 2020 to January 2021.

#### 2.3.2. Depression

The Thai version of the Patient Health Questionnaire for Adolescents (PHQ-A), a self-reporting questionnaire, consists of nine questions to describe the severity of depression symptoms over the past two weeks, rated from 0 (did not apply to me at all) to 3 (applied to me almost every day). The internal consistency coefficient of the Thai version of the PHQ-A was reported as 0.92 [35] in a previous study; it was 0.806 among the senior high school students who participated in this study.

#### 2.3.3. Achievement

The learning achievements of students in this study were estimated using students’ grade point averages (GPA).

#### 2.3.4. Anxiety

The Screen for Child Anxiety Related Disorders (SCARED-Thai version), which consists of 41 questions evaluating participants’ feelings over the past three months, initially determined anxiety disorders, including panic disorder and social anxiety disorder. The SCARED-Thai version had excellent reliability, with an internal consistency of 0.913 for the SCARED-Child and 0.925 for the SCARED-Parent forms [36]. Its reliability among senior high school students was 0.934.

#### 2.3.5. Interpersonal Relationships

Interpersonal relationships, measured by the average points for 6 items, evaluated the respondents’ relationships with other people (i.e., fathers, mothers, caregivers, siblings, teachers, and peers). Points for each item ranged from 0 (poor relationship) to 10 (excellent relationship). The reliability of this scale was 0.623.

#### 2.3.6. Self-Esteem

The Rosenberg Self-Esteem Scale, a 4 Likert-scale questionnaire (1 = strongly disagree, 2 = disagree, 3 = agree, and 4 = strongly agree) with ten items, was used to evaluate the participants’ self-esteem. A previous study of Thai students using the Thai version of this scale found that the internal consistency coefficient was 0.86 [37]; it was 0.719 among the senior high school students in this study.

#### 2.3.7. Life Satisfaction

Life satisfaction was self-evaluated using a visual analogue scale ranging from 0 (no satisfaction) to 10 (very satisfied).

### 2.4. Statistical Analysis

Initially, descriptive statistics were applied to determine the central tendency and data dispersion of each variable, as well as to check the normality of the skewness (normal value = ±2) and the kurtosis (normal value = ±10). Additionally, analysis of the VIF and tolerance to illustrate the relationships among variables was applied to determine the appropriate analytical methods for the variables influencing student achievement and depression. The VIF < 10 and tolerance (1/VIF) > 1/10 were considered as a lack of multicollinearity [38].

Finally, the validity of the model of academic achievement and five psychological factors, including depression (DEP), anxiety (ANX), self-esteem (SEL), life satisfaction (LIF), and interpersonal relationships (INT), were examined. The missing values of these variables were replaced with their means. All variables were included in the analysis to determine all plausible direct and indirect effect paths using path analysis (PA) with a maximum likelihood estimator. According to the PA, the goodness of fit statistics were determined by the relative chi-square (χ^2^/df), which was no more than two [39] and had a *p*-value of more than 0.05 [40]. Additionally, the comparative fit index (CFI) and the Tucker-Lewis index (TLI) were greater than 0.95, while the root mean square error index of approximation (RMSEA) and standardized root mean square residual (SRMR) were less than 0.05 [41]. All analyses were performed using Mplus 7.4 and Stata 17.

## 3. Results

Of 315 students who participated in the survey, 30 students did not complete the questionnaires. Thus, only 285 participants (90.48%) were included in the data analysis. Most students were female (female = 56.84%, male = 43.16%), aged 14–19 years, and studied in grades 10 and 11. Most of their parents included both their fathers and mothers (64.44%). Most students lived with their father and mother in their parents’ house (81.05%). More than half of the students (58.80%) attended tutorial courses; 22.80% attended more than five subjects, and 9.94% attended class for more than 20 h a week. Most students decided to study tutorial courses by themselves. Most students received an allowance of 101–500 (48.42%) or less than 100 (47.02%) Thai baht (THB) per day. Some students (19.43%) reported knowing their family debt status. Twenty-six students (9.12%) availed of the educational loan program offered by the government (Table 2).

The means and standard deviations of academic achievement (ACH), depression scores (DEP), anxiety scores (ANX), self-esteem scores (SEL), life satisfaction scores (LIF), and interpersonal relationships scores (INT) were 3.446 (0.460), 7.601 (4.276), 24.513 (13.331), 28.629 (4.285), 6.783 (2.067), and 7.881 (1.687), respectively. According to the model assumptions, the examined outcomes reveal that (1) the overall distribution trend of the data was normal (the skewness was between ±2 and the kurtosis was between ±10); (2) tolerance of variables ranged from 0.572 (DEP)–0.911 (ACH), which was close to 1; and (3) VIFs ranged from 1.098 (ACH)–1.748 (DEP), which did not exceed 10. These values indicate that the overall data did not violate the assumptions. Considering the relationships between the studied variables in the hypothetical model, 11 out of 15 pairs of variables were significantly different from zero, which were classified as five pairs of positive correlations and six pairs of negative correlations (Table 3).

The path analysis results show that all fit indices of the hypothetical model are harmonious with the empirical data (Chi-square (χ^2^) = 0.295 (df = 1), relative Chi-square (χ^2^/df) = 0.295, RMSEA = 0.000, CFI = 1.000, TLI = 1.000, SRMR = 0.005).

When assessing the influence of five variables (DEP, ANX, SEL, LIF, INT) on academic achievement, anxiety (*β* = 0.375) had a significant positive direct effect, whereas depression (*β* = −0.216) had a negative direct effect. Anxiety also had a negative indirect effect on achievement (*β* = −0.076). Interpersonal relationships were not significantly associated with achievement; however, a significant indirect effect was noted (*β* = 0.062). The total indirect effects of self-esteem and life satisfaction did not significantly influence achievement; however, there were some significant effects in specific paths through depression and anxiety.

According to the estimation of the four influence variables (ANX, INT, LIF and SEL), anxiety (*β* = 0.353) had a direct positive effect on depression. However, self-esteem (*β* = −0.290), life satisfaction (*β* = −0.131), and interpersonal relationships (*β* = −0.104) had negative indirect effects on depression. Additionally, self-esteem (*β* = −0.054), life satisfaction (*β* = −0.218), and interpersonal relationships (*β* = −0.214) had negative indirect effects on depression.

Self-esteem had a negative direct effect on anxiety (*β* = −0.364). Although there were no direct effects of life satisfaction and interpersonal relationships on anxiety, negative indirect effects were present for both of these factors. (*β* = −0.151 and −0.187, respectively).

Considering the standardized total effect coefficients, only depression (*β* = −0.216) and anxiety (*β* = 0.299) influenced academic achievement. Anxiety had a positive effect (*β* = 0.353), whereas self-esteem (*β* = −0.301), life satisfaction (*β* = −0.321), and interpersonal relationships (*β* = −0.309) had negative effects on depression. Self-esteem (*β* = −0.364), life satisfaction (*β* = −0.274), and interpersonal relationships (*β* = −0.139) also had negative effects on anxiety (Table 4 and Figure 1).

## 4. Discussion

According to the validation of the factor model influencing academic achievement, the overall outcomes show that our hypothetical model is consistent with the empirical data. The variables studied based on previous findings confirm the causal relationships of depression and anxiety on student achievement. All factors illustrate both direct and indirect effects. However, the effect of anxiety was contrary to our hypothesis. The present PA indicates that anxiety has a significant positive direct effect on academic achievement. Although self-esteem and interpersonal relationships seem to have positive direct effects on academic achievement, these did not reach statistical significance. Based on the present model structure, interpersonal relationships, life satisfaction, and self-esteem remain important factors in explaining the variability of depression and anxiety. According to the overall effects, each factor’s interplay can account for variation in academic achievement, depression, and anxiety of 11.60%, 42.80%, and 17.60%, respectively.

### 4.1. Academic Achievement

Depression and academic achievement might have a bidirectional relationship. According to the present study, depression is associated with decreased academic achievement. As known, low academic achievement can induce depressive symptoms in some students. A previous study found that depression had a negative impact on academic achievement [3]. Deterioration of learning ability may be caused by depressive symptoms, including loss of interest, difficulty sleeping, slowed thought processes, fatigue, inappropriate guilt, and difficulty concentrating and thinking [42]. Conversely, students with low academic achievement often have low self-esteem, possibly leading to the development of depression. Given the high prevalence of depression in students, an appropriate program to reduce the risk of depressive occurrence and reduce students’ expectations with regard to appropriate acceptance of academic achievement may be useful.

Our findings show that anxiety also plays an important role in influencing academic achievement. The previous statistical study of academic achievement found the inverted U-shaped relationship of anxiety more predictive of the students’ academic achievement than a straight-line relationship [10]. Based on this evidence, students with an appropriate level of anxiety (not too much or too little) had the highest achievements. Additional evidence also illustrates the effect of anxiety on motivation and success [11]. The findings show that 49% of high-achieving students had moderate anxiety, and 41% had severe anxiety. A high level of internal and external motivation was found in students with moderate situational anxiety, while low motivation was found in those students with high levels of latent anxiety. Although higher levels of anxiety increase academic success, they can decrease learning motivation. Low anxiety levels reduce learning ability but increase motivation. Accordingly, a moderate level of anxiety can be optimally helpful in enhancing academic achievement.

Self-esteem was not related to academic achievement in the present study. This is in contrast with previous studies, which suggest a statistically significant positive correlation between academic achievement and self-esteem [19,20,21]. Interestingly, our results show the significant indirect effect of self-esteem in some paths considered for depression and anxiety. Depression and anxiety were independently associated with self-esteem; this was possibly a result of the mediation effect of depression and/or anxiety on the relationship between self-esteem and academic achievement. Moreover, the significant indirect effect of life satisfaction is also noted in some paths, along with self-esteem, depression, and anxiety. The analysis of the mediation effects of these factors is noteworthy for further study. In addition, significant indirect effects might result from measurement error. A structural equation model that determines the latent variables and accounts for measurement error should be conducted to confirm this finding.

Although there was no direct effect of interpersonal relationships on academic achievement, the present findings illustrate that interpersonal relationships have an indirect effect on student achievement. This is possibly explained by the way that good relationships with others, which are essential for adolescent development, may facilitate those adolescents’ access to academic resources through closed and trusted relationships [43]. This is also compatible with previous studies which illustrate the positive relationship between the home environment and the academic achievement [19]. An additional study also found that peer groups have a significant influence on students’ academic achievements [15], since characteristics of each peer group impact their motivation to achieve academically [16,17,18].

### 4.2. Depression and Anxiety

There is a relationship between anxiety and depression. Previous studies illustrated that depression is associated with stress and anxiety, while anxiety, sleep quality, and stress are the crucial factors predicting the occurrence of depression [2]. Similar to previous studies, our findings indicate that anxiety has a positive direct effect on depression. Again, stress management training for anxious students may help them to reduce anxiety, which may decrease the incidence of depression.

Self-esteem was only one factor that significantly influenced both depression and anxiety. Typically, low self-esteem in adolescents is one of the risk factors for development of depression. Previous studies found that self-esteem was negatively associated with the incidence of anxiety and depression. Since anxiety can cause depression, low self-esteem also indirectly affects the incidence of depression in adolescents by increasing anxiety [14]. This evidence is compatible with the present study which illustrates the direct, indirect, and overall effect of self-esteem on depression and anxiety. A program promoting self-esteem with positive reinforcement for students may reduce anxiety and depression and enhance students’ academic achievement.

According to interpersonal theory, depression is related to individual interactions with others as eliciting rejection [44]. The first onset of depression often occurs when an individual encounters deleterious interpersonal experiences [45]. The previous study presented evidence which supports our findings regarding the negative direct, indirect, and overall effects of interpersonal relationships on depression. Hence, encouraging students’ interpersonal relationships may prevent or treat depression in students. Since social skills are evidently positively related to interpersonal relationships, social skills training is useful for students with a high risk of depression.

Life satisfaction is generally related to mental health. Previous studies illustrated the relationship between life satisfaction and depression in youth [46]. Seo et al. [26] found that life satisfaction was negatively associated with depression and had an indirect effect on reducing depression. Since life satisfaction is positively associated with self-esteem [27], high life satisfaction possibly decreases the incidence of depression. This evidence is compatible with our present model, which shows that life satisfaction has a negative direct, indirect, and overall influence on depression in students. Therefore, prevention strategies, including encouraging students to attend interesting activities, may enhance their life satisfaction and decrease the incidence of depression.

### 4.3. Limitations

There were some limitations in this study. Firstly, this research had a limited sample size. Thus, the findings should be used cautiously. Secondly, samples were only retrieved from urban high school students; their characteristics might differ from rural high school students. Therefore, generalization is possibly not appropriate. Finally, as this study focused on only five factors, it may not be comprehensive. Additional factors such as parental support, learner characteristics, learning style, attitude, achievement motivation, self-discipline, self-confidence, and self-efficacy should be included in a future study. In addition, some factors might have reverse causality, such as the reciprocal relationship between self-esteem and academic achievement (students’ higher academic achievement may influence their positive self-views). However, we did not investigate reverse causality in our analyses due to the limitation of the cross-sectional design, which does not allow an investigation of reverse causality.

## 5. Conclusions

According to the present findings, anxiety has a positive direct effect on academic achievement, while interpersonal relationships have a positive indirect effect on academic achievement. Additionally, anxiety had a positive direct effect on depression, while self-esteem, interpersonal relationships, and life satisfaction had negative direct effects on depression. Since anxiety can enhance academic achievement and increase the incidence of depression, a program that stimulates the optimal and appropriate level of anxiety may be useful. Although anxiety appears positively related to academic achievement, it relatively influences the incidence of depression. Thus, encouraging self-esteem, interpersonal relationships, and life satisfaction in students can promote academic ability and decrease the risk of depression and anxiety. Further well-designed and large sample-size studies should be conducted to confirm these findings.

## Figures and Tables

**Figure 1 healthcare-10-01163-f001:**
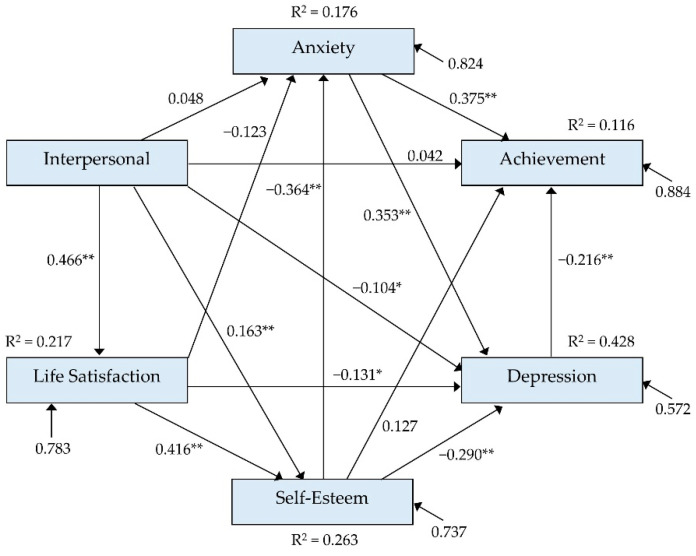
The standardized regression coefficient of factors influencing achievement and depression; * *p* < 0.05, ** *p* < 0.01, R^2^: coefficient of determination.

**Table 1 healthcare-10-01163-t001:** Summarized results of the synthesis of factors influencing students’ achievement and depression.

Dependent Variable	Independent Variables	Hypotheses
Achievement	Depression	**(H1).** Negative effect of depression on achievement
	Anxiety	**(H2).** Negative effect of anxiety on achievement
	Interpersonal relationships	**(H3).** Positive effect of interpersonal relationships on achievement
	Self-esteem	**(H4).** Positive effect of self-esteem on achievement
Depression	Anxiety	**(H5).** Positive effect of anxiety on depression
	Interpersonal relationships	**(H6).** Negative effect of interpersonal relationships on depression
	Life satisfaction	**(H7).** Negative effect of life satisfaction on depression
	Self-esteem	**(H8).** Negative effect of self-esteem on depression
Anxiety	Interpersonal relationships	**(H9).** Negative effect of interpersonal relationships on anxiety
	Life satisfaction	**(H10).** Negative effect of life satisfaction on anxiety
	Self-esteem	**(H11).** Negative effect of self-esteem on anxiety
Self-esteem	Interpersonal relationships	**(H12).** Positive effect of interpersonal relationships on self-esteem
	Life satisfaction	**(H13).** Positive effect of life satisfaction on self-esteem
Life Satisfaction	Interpersonal relationships	**(H14).** Positive effect of interpersonal relationships on life satisfaction

**Table 2 healthcare-10-01163-t002:** Demographic data.

Demographics	Frequencies	Percentages
**Gender**		
Male	123	43.16
Female	162	56.84
**Age**		
14	3	1.06
15	63	22.34
16	98	33.69
17	91	32.27
18	28	9.93
19	2	0.71
**Type of school**		
Private school	112	39.30
Government school	133	46.66
Demonstration school	40	14.04
**Grade**		
10	109	38.24
11	99	34.74
12	77	27.02
**Primary parent**(**s**)		
Father and Mother	183	64.44
Father	12	4.23
Mother	63	22.18
Cousin	26	9.15
**Living with**		
Father and Mother	168	59.36
Father	10	3.54
Mother	51	18.02
Cousin	33	11.66
Friend	6	2.12
Alone	15	5.30
**Residency**		
House	231	81.05
Dormitory	54	18.95
**Educational loan**		
No	259	90.88
Yes	26	9.12
**Household debt**		
No	94	33.22
Yes	55	19.43
Unknown	134	47.35
**Students’ daily allowance (THB)** ^a^		
<100	134	47.02
101–500	138	48.42
501–1000	7	2.46
>1000	6	2.10
**Tutorial class attending**		
No	117	41.20
Yes	167	58.80
**Number of subjects of tutorial class**		
1–2	87	50.88
3–4	45	26.32
>5	39	22.80
**Period of tutorial class (hours)**		
<10	125	73.10
11–20	29	16.96
21–30	12	7.02
>30	5	2.92
**Reason for attending tutorial class**		
Own decision	156	91.23
Suggested by friends	4	2.34
Forced by parent(s)	5	2.92
Other	6	3.51

^a^ Note that 1 USD is approximately 34.10 THB (Thai baht).

**Table 3 healthcare-10-01163-t003:** Variables influencing depression and learning achievement of high school students.

	ACH	DEP	ANX	SEL	LIF	INT
**Achievement: ACH**	1.000					
**Depression: DEP**	−0.102	1.000				
**Anxiety: ANX**	0.204 **	0.522 **	1.000			
**Self-esteem: SEF**	0.105	−0.535 **	−0.407 **	1.000		
**Life Satisfaction: LIF**	0.044	−0.421 **	−0.279 **	0.492 **	1.000	
**Interpersonal relationships: INT**	0.104	−0.317 **	−0.139 *	0.357 **	0.466 **	1.000
Mean	3.446	7.601	24.513	28.629	6.783	7.881
SD	0.460	4.276	13.331	4.285	2.067	1.687
Skewness	−0.825	0.615	0.542	−0.024	−0.505	0.468
Kurtosis	0.272	0.535	0.334	0.091	0.068	3.805
Tolerance	0.911	0.572	0.699	0.601	0.641	0.749
VIF	1.098	1.748	1.431	1.665	1.560	1.335

* *p* < 0.05, ** *p* < 0.001.

**Table 4 healthcare-10-01163-t004:** Regression coefficients of factors associated with achievement, depression, and anxiety.

Path Directions	Direct Effect		Indirect Effect		Total Effect	
	*b*	*β*	*b*	*β*	*b*	*β*
**INT → ACH**	0.011	0.042	0.017 *	0.062 *	0.028	0.104
INT → DEP → ACH	-	-	0.006	0.022	-	-
INT → ANX → ACH	-	-	0.005	0.018	-	-
INT → SEL → ACH	-	-	0.006	0.021	-	-
INT → ANX → DEP → ACH	-	-	−0.001	−0.004	-	-
INT → LIF → DEP → ACH	-	-	0.004	0.013	-	-
INT → SEL → DEP → ACH	-	-	0.003	0.010	-	-
INT → LIF → ANX → ACH	-	-	−0.006	−0.021	-	-
INT → SEL → ANX → ACH	-	-	−0.006 *	−0.022 *	-	-
INT → LIF → SEL → ACH	-	-	0.007	0.025	-	-
INT → LIF → ANX → DEP → ACH	-	-	0.001	0.004	-	-
INT → SEL → ANX → DEP → ACH	-	-	0.001	0.005	-	-
INT → LIF → SEL → DEP → ACH	-	-	0.003 *	0.012*	-	-
INT → LIF → SEL → ANX → ACH	-	-	−0.007 **	−0.026 **	-	-
INT → LIF → SEL → ANX → DEP → ACH	-	-	0.001 *	0.005 *	-	-
**LIF** → **ACH**	-	-	0.006	0.025	0.006	0.025
LIF → DEP → ACH	-	-	0.006	0.028	-	-
LIF → ANX → ACH	-	-	−0.010	−0.046	-	-
LIF → SEL → ACH	-	-	0.012	0.053	-	-
LIF → ANX → DEP → ACH	-	-	0.002	0.009	-	-
LIF → SEL → DEP → ACH	-	-	0.006 *	0.026 *	-	-
LIF → SEL → ANX → ACH	-	-	−0.013 **	−0.057 **	-	-
LIF → SEL → ANX → DEP → ACH	-	-	0.003 *	0.012 *	-	-
**SEL** → **ACH**	0.014	0.127	−0.005	−0.046	0.009	0.081
SEL → DEP → ACH	-	-	0.007 *	0.063 **	-	-
SEL → ANX → ACH	-	-	−0.015 **	−0.136 **	-	-
SEL → ANX → DEP → ACH	-	-	0.003 *	0.028 *		
**ANX** → **ACH**	0.013 **	0.375 **	−0.003 **	−0.076 **	0.010 **	0.299 **
ANX → DEP → ACH	-	-	−0.003 **	−0.076 **	-	-
**DEP** → **ACH**	−0.023 **	−0.216 **	-	-	−0.023 **	−0.216 **
**INT** → **DEP**	−0.264 *	−0.104 *	−0.541 **	−0.214 **	−0.805 **	−0.317 **
INT → ANX → DEP	-	-	0.043	0.017	-	-
INT → LIF → DEP	-	-	−0.154 *	−0.061 *	-	-
INT → SEL → DEP	-	-	−0.120 *	−0.047 *	-	-
INT → LIF → ANX → DEP	-	-	−0.051	−0.020	-	-
INT → SEL → ANX → DEP	-	-	−0.053 *	−0.021 *	-	-
INT → LIF → SEL → DEP	-	-	−0.143 **	−0.056 **	-	-
INT → LIF → SEL → ANX → DEP	-	-	−0.063 **	−0.025 **	-	-
**LIF** → **DEP**	−0.270 *	−0.131 *	−0.450 **	−0.218 **	-0.720 **	-0.348 **
LIF → ANX → DEP	-	-	−0.090	−0.043	-	-
LIF → SEL → DEP	-	-	−0.250 **	−0.121 **	-	-
LIF → SEL → ANX → DEP	-	-	−0.111 **	−0.054 **	-	-
**SEL** → **DEP**	−0.290 **	−0.290 **	−0.128 **	−0.129 *	−0.418 **	−0.419 **
SEL → ANX → DEP	-	-	−0.128 **	−0.129 **	-	-
**ANX** → **DEP**	0.113 **	0.353 **	-	-	0.113 *	0.353 *
**INT** → **ANX**	0.378	0.048	−1.478 **	−0.187 **	−1.100 *	−0.139 *
INT → LIF → ANX	-	-	−0.451	−0.057	-	-
INT → SEL → ANX	-	-	−0.468 *	−0.059 *	-	-
INT → LIF → SEL → ANX	-	-	−0.558 **	−0.071 **	-	-
**LIF** → **ANX**	−0.791	−0.123	−0.977 **	−0.151 **	−1.768 **	−0.274 **
LIF → SEL → ANX	-	-	−0.977 **	−0.151 **	-	-
**SEL** → **ANX**	−1.132 **	−0.364 **	-	-	−1.132 **	−0.364 **

** *p* < 0.01, * *p* < 0.05. ACH, achievement; ANX, anxiety; DEP, depression; INT, interpersonal relationships; LIF, life satisfaction; SEL, self-esteem.

## Data Availability

The datasets analyzed in this study are available from the corresponding author on reasonable request.

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
