# Peer review of "Psychological Factors Influencing Achievement of Senior High School Students"

_healthcare, 2022, doi:10.3390/healthcare10071163_

Round 1

Reviewer 1 Report

Self-esteem and mental health in adolescence are important variables that have been related to healthy or risky experiences in this important period of development. Self-esteem is understood as a set of feelings and thoughts of the individual in relation to their own value, competence, confidence, adequacy and ability to face challenges, which reflects on a positive or negative attitude towards themselves. It is considered an important factor that influences the way a person perceives, feels and responds to the world. High or low self-esteem is related to the person's experiences throughout life, especially those related to affection, love, appreciation, success or failure. It is important to consider that adolescence is a period of development in which physical, cognitive and social changes occur. Adolescents start to play another role in front of their family, being more participatory and showing greater autonomy. Therefore, emotional integrity becomes important, as healthy levels of self-esteem and mental health can drive a more adaptive performance to the demands of the environment during this period. Currently, self-esteem is seen as one of the fundamental social indicators for understanding personal growth and progress. It is a decisive point to be observed in a healthy and complete personal development in its multiple areas. This research was very well conducted and its results reinforce all these concepts above that are of paramount importance for youth education.

I would like to know more about how the socioeconomic status cited in the Demographic data was evaluated. Since there were participants from different types of schools: private, government and demonstration high schools.

I also felt that I could have a table indicating the Patient Health Questionnaire for Adolescents (PHQ-A), SCARED-Thai version and Rosenberg's Self-Esteem Scale values for this sample (as in table 2).

Author Response

Response to Reviewer 1 Comments

Self-esteem and mental health in adolescence are important variables that have been related to healthy or risky experiences in this important period of development. Self-esteem is understood as a set of feelings and thoughts of the individual in relation to their own value, competence, confidence, adequacy and ability to face challenges, which reflects on a positive or negative attitude towards themselves. It is considered an important factor that influences the way a person perceives, feels and responds to the world. High or low self-esteem is related to the person's experiences throughout life, especially those related to affection, love, appreciation, success or failure. It is important to consider that adolescence is a period of development in which physical, cognitive and social changes occur. Adolescents start to play another role in front of their family, being more participatory and showing greater autonomy. Therefore, emotional integrity becomes important, as healthy levels of self-esteem and mental health can drive a more adaptive performance to the demands of the environment during this period. Currently, self-esteem is seen as one of the fundamental social indicators for understanding personal growth and progress. It is a decisive point to be observed in a healthy and complete personal development in its multiple areas. This research was very well conducted and its results reinforce all these concepts above that are of paramount importance for youth education.

Point 1: I would like to know more about how the socioeconomic status cited in the Demographic data was evaluated. Since there were participants from different types of schools: private, government and demonstration high schools.

Response 1: Thank you for your suggestion. We have added the demographic information, including educational loan, household debt, and daily allowance, in Table 2 and the Results section accordingly [Page 5, Lines 215 – 225].

“Of 315 students were participated in the survey, 30 participants did not complete the questionnaires. Thus, only 285 participants (90.48%) were included in the data analysis. Most students were female (female=56.84%, male=43.16%), aged 14-19 years, studied in grade 10 and 11. Most of their parents were their father and mother (64.44%). Most students lived with their father and mother in their parents’ house (81.05%). More than half of the students (58.80%) attended the tutorial courses, 22.80% attended more than five subjects, and 9.94% attended the class for more than 20 hours a week. Most of the students decided to study tutorial courses by themselves. Most students received an allowance of about 101 – 500 (48.42%) and less than 100 (47.02%) Thai baht per day. Some students (19.43%) reported knowing about their family debt status. Twenty-six students (9.12%) attended the educational loan program offered by the government (Table 2).”

Table 2. Demographic data.

Demographics

Frequencies

Percentages

Gender            

                  Male

123

43.16

                  Female

162

56.84

Age

                  14

3

1.06

                  15

63

22.34

                  16

98

33.69

                  17

91

32.27

                  18

28

9.93

                  19

2

0.71

Type of school               

                  Private school

112

39.30

                  Government school

133

46.66

                  Demonstration school

40

14.04

Grade               

                  10

109

38.24

                  11

99

34.74

                  12

77

27.02

Primary parent(s)

                  Father & Mother

183

64.44

                  Father

12

4.23

                  Mother

63

22.18

                  Cousin

26

9.15

Living with          

                  Father & Mother

168

59.36

                  Father

10

3.54

                  Mother

51

18.02

                  Cousin

33

11.66

                  Friend

6

2.12

                  Alone

15

5.30

Residency    

                  House

231

81.05

                  Dormitory

54

18.95

Educational loan  

                  No

259

90.88

                  Yes

26

9.12

Household debt  

                  No

94

33.22

                  Yes

55

19.43

                  Unknown

134

47.35

Students’ daily allowance (Thai baht)a

                  < 100

134

47.02

                  101 – 500

138

48.42

                  501 – 1,000

7

2.46

                  > 1,000

6

2.10

Tutorial class attending

                  No

117

41.20

                  Yes

167

58.80

Number of subjects of tutorial class

                  1-2

87

50.88

                  3-4

45

26.32

                  > 5

39

22.80

Period of tutorial class (hours)

                  < 10

125

73.10

                  11-20

29

16.96

                  21-30

12

7.02

                  > 30

5

2.92

Reason for attending tutorial class

                  Own decision

156

91.23

                  Suggested by friends

4

2.34

                  Forced by parent(s)

5

2.92

                  Others

6

3.51

              a 1 USD is approximately 34.10 Thai baht

Point 2: I also felt that I could have a table indicating the Patient Health Questionnaire for Adolescents (PHQ-A), SCARED-Thai version and Rosenberg's Self-Esteem Scale values for this sample (as in table 2).

Response 2: Thank you for your suggestion. The descriptive statistics (i.e., means, standard deviations, related psychometric measurement (PHQ-A, SCARED-Thai version, Rosenberg’s self-esteem, and interpersonal relationship) of the study samples were separately presented in Table 3 and the Results section [Page 7, Lines 231 – 242]. However, we have corrected some values in the table according to the correctness of using data.

              “The means and standard deviations of academic achievement (ACH), depression score (DEP), anxiety score (ANX), self-esteem score (SEL), life satisfaction score (LIF), and interpersonal relationship (INT) were 3.446 (0.460), 7.601 (4.276), 24.513 (13.331), 28.629 (4.285), 6.783 (2.067), and 7.881 (1.687), respectively. According to the model assumptions, the present examined outcomes revealed that 1) the overall distribution trend of the data was normal (the skewness was between ±2 and the kurtosis was between ±10), 2) Tolerance of each variable was between 0.572 (DEP) – 0.911 (ACH), which was close to 1, and 3) VIF was between 1.098 (ACH) – 1.748 (DEP), which did not exceed 10. These values indicated that the overall data did not violate the assumptions. Considering the relationship between the studied variables in the hypothetical model, 11 of 15 pairs of variables were significantly different from zero, which were classified as five pairs of positive correlations and six pairs of negative correlations (Table 3).”

Table 3. Variables influencing depression and learning achievement of high school students.

ACH

DEP

ANX

SEL

LIF

INT

Achievement: ACH

1.000

Depression: DEP

-0.102

1.000

Anxiety: ANX

0.204**

0.522**

1.000

Self-Esteem: SEF

0.105

-0.535**

-0.407**

1.000

Life Satisfy: LIF

0.044

-0.421**

-0.279**

0.492**

1.000

Interpersonal relationship: INT

0.104

-0.317**

-0.139*

0.357**

0.466**

1.000

Mean

3.446

7.601

24.513

28.629

6.783

7.881

SD

0.460

4.276

13.331

4.285

2.067

1.687

Skewness

-0.825

0.615

0.542

-0.024

-0.505

0.468

Kurtosis

0.272

0.535

0.334

0.091

0.068

3.805

Tolerance

0.911

0.572

0.699

0.601

0.641

0.749

VIF

1.098

1.748

1.431

1.665

1.560

1.335

*p<0.05, ** p<0.001.

Reviewer 2 Report

Dear authors,

Many factors have been identified as influencing academic performance and depression in secondary school students, but not all possible cause-and-effect relationships are yet clear. In order to show the direct and indirect effects of different influences, a path analysis was conducted in this study to examine the relationship between academic achievement, interpersonal relationships, life satisfaction, self-esteem, anxiety and depression among upper secondary school students. It is an interesting work, but I would like to suggest some corrections to the authors

Introduction:
It is poorly structured, with authors moving from one factor to another without a flowing link, and sometimes it is not clear what they are trying to say (e.g.: However....). I would also advise a brief definition of the terms used as variables, in a logical sequence. 
Methodology - I would advise only a written description of the links without Figure 1
Results: Please remove the references in Table 1, this is very opaque and misleading.
We usually start the discussion with the most important findings of the research. Hypothesis-based characterization is less desirable for a paper.

Conclusion: solid

Author Response

Response to Reviewer 2 Comments

Dear authors,

Many factors have been identified as influencing academic performance and depression in secondary school students, but not all possible cause-and-effect relationships are yet clear. In order to show the direct and indirect effects of different influences, a path analysis was conducted in this study to examine the relationship between academic achievement, interpersonal relationships, life satisfaction, self-esteem, anxiety and depression among upper secondary school students. It is an interesting work, but I would like to suggest some corrections to the authors.

Point 1: Introduction - It is poorly structured, with authors moving from one factor to another without a flowing link, and sometimes it is not clear what they are trying to say (e.g.: However....). I would also advise a brief definition of the terms used as variables, in a logical sequence. 

Response 1: Thank you for your suggestion. We have revised the structure of the content in the Introduction section to be connected and clearer. We have moved the literature review for each studied factor to the “1.1 Literature review” subsection [Page 1, Lines 33 – 44, Page 2, Lines 45 – 97, and Page 3, Lines 98 – 102] as follows.

1. Introduction

Academic achievement usually relies on individual intellectual and cognitive ability. However, many students encounter problems in learning performance. In addition to the efficacy of educational programs, students' non-academic attributes, including physical and mental health, could also influence learning performance [1]. Depression, a high prevalence among the students, could decrease academic achievement [2]. To our knowledge, the causal relationship between achievement and depression has been displayed in different dimensions [3-6]. Anxiety is one factor that plays an essential role in both academic achievement [3,6-11] and depression [2,9,12-14]. In addition to depression and anxiety, previous evidence revealed that several psychological factors were also associated with academic achievement, including interpersonal relationship [15-19] and self-esteem [19-21].

Previous evidence also illustrates the significant association between interpersonal relationship [13,22-25], self-esteem [13,14], and life satisfaction [26,27] toward depression and anxiety. Moreover, the inter-correlation between each factor was proposed in several studies. Interpersonal relationship was associated with self-esteem [13,14,19,28], and life satisfaction [29,30]. Meanwhile, the influences of life satisfaction on anxiety [26] and self-esteem [27] were presented in the previous study. Additionally, self-esteem was also associated with anxiety [14].

The overlapping associations of self-esteem, life satisfaction, and interpersonal relationship to depression and anxiety may be mediators contributing to their influences on academic achievement. Previous studies which focused on finding factors that directly influence academic achievement may lead to the neglect of some indirect associations. Therefore, the aim of this study was to illustrate the association between study achievement and the potential influence of psychological factors among high school students using path analysis which allowed to investigate of both direct and indirect effects of each factor.

1.1 Literature review

According to previous studies, we found the association of study variables toward academic achievement and each other in the different dimensions. The authors conducted a literature review for the association of studied factors on the achievement and each other revealed in previous studies for the hypothetical model in this study.

Depression is a mood disorder that causes a persistent feeling of sadness and loss of interest. It often presents with physical symptoms, primarily fatigue, pain, or sleep disturbance. The depressed mood may or may not be present. Depression could play an essential role in affecting academic achievement. A previous study among secondary school students in Iran found that students with a high level of depression had a low academic achievement [3]. Several studies also showed that depression and academic performance correlated significantly [4-6,31]. Interestingly, the depression also showed a positive relationship with academic performance among the students with low social support in a previous study [5].

Anxiety has shown both positive and negative effects on academic achievement. An appropriate level of anxiety positively enhances work efficiency, including learning ability, while a high level may negatively affect academic achievement [9,22]. A previous study found that students with low levels of academic achievement tend to have higher levels of anxiety, depression, and stress [6,12]. Additionally, a previous study suggested that students who achieved low levels of academic achievement tend to have higher levels of anxiety, depression, and stress [6].

Interpersonal relationship in adolescents is one interesting exogenous variable related to mental health and academic achievement. Previous evidence illustrated that adolescents with interpersonal relationship difficulty have a high risk of depression. Meanwhile, positive childhood experiences such as being part of the school, closed relationships with peers, and good family support could prevent them from depression and anxiety [13,22,23]. Additionally, good relationships among family members increase academic achievement in adolescents [19]. Previous studies also suggested that peer relationship is significantly related to academic achievement [15-18].

            Life satisfaction also plays a role related to depression. A study on college students found that life satisfaction was negatively associated with depression [26]. Interestingly, self-esteem was positively correlated with life satisfaction [27].

Generally, self-esteem is the attitude of a person who is satisfied or dissatisfied with himself. Maslow classified esteem needs into two categories: i) esteem for oneself is an internal quality, including dignity, achievement, mastery, independence, and ii) the desire for reputation or respect from others. Maslow indicated that the need for respect or reputation is most important for children and adolescents and precedes genuine self-esteem or dignity [32]. Self-esteem was also negatively correlated with the incidence of depression, anxiety, and suicidal ideation [13,14]. Adolescents with high self-esteem are less likely to experience depression than those with low self-esteem. A previous study of high school students found that educational stress and domestic violence are risk factors for low self-esteem [14]. Additionally, academic achievement and self-esteem were highly positively correlated [33]. Additional studies illustrate a significant positive correlation between self-esteem and academic achievement. The high self-esteem increases academic achievement in those students [19,20].”

Point 2: Methodology - I would advise only a written description of the links without Figure 1

Response 2: Thank you for your suggestion. We have removed Figure 1 from the manuscript and revised the related paragraph to describe the hypothesized links of factors in the Materials and Methods section [Page 3, Lines 104 – 119] as follows.

2.1. Hypotheses

According to the synthesis of documents and related research, the factors influencing academic achievement among high school students which considered to be included in this study were depression, anxiety, self-esteem, life satisfaction, and interpersonal. We hypothesized that the negative influence factors for academic achievement (depression (H1) and anxiety (H2)) and positive influence factors (interpersonal (H3) and self-esteem (H4)). Additionally, we also hypothesized that the positive influence of anxiety (H5) and the negative influence factors for depression included interpersonal relationship (H6), life satisfaction (H7), and self-esteem (H8) for depression. The negative influence factors for anxiety included interpersonal relationship (H9), life satisfaction (H10), and self-esteem (H11) were also determined. Since previous evidence illustrated the inter-correlation among those factors, the authors additionally hypothesized that the positive influence of interpersonal relationship (H12) and life satisfaction (H13) on self-esteem and the positive influence factor interpersonal relationship (H14) on life satisfaction (Table 1). We aimed to examine both direct effects in all 14 hypothesized pathways and all plausible indirect effects between each factor and other mediators.”

Point 3: Results: Please remove the references in Table 1, this is very opaque and misleading.

We usually start the discussion with the most important findings of the research. Hypothesis-based characterization is less desirable for a paper.

Response 3: Thank you for your suggestion. We have removed the references from Table 1 and cited them only in the Hypotheses subsection, Materials and Methods section accordingly.

Table 1. Summarized results of the synthesis of factors influencing students' achievement and depression.

Dependent variable

Independent Variables

Hypotheses

Achievement

Depression

(H1) Negative effect of depression on achievement

Anxiety

(H2) Negative effect of anxiety on achievement

Interpersonal relationship

(H3) Positive effect of interpersonal relationship on achievement

Self-esteem

(H4) Positive effect of self-esteem esteem on achievement

Depression

Anxiety

(H5) Positive effect of anxiety on depression

Interpersonal relationship

(H6) Negative effect of interpersonal relationship on depression

Life satisfaction

(H7) Negative effect of life satisfaction on depression

Self-esteem

(H8) Negative effect of self-esteem on depression

Anxiety

Interpersonal relationship

(H9) Negative effect of interpersonal relationship on anxiety

Life satisfaction

(H10) Negative effect of life satisfaction on anxiety

Self-esteem

(H11) Negative effect of self-esteem on anxiety

Self-esteem

Interpersonal relationship

(H12) Positive effect of interpersonal relationship on self-esteem

Life satisfaction

(H13) Positive effect of life satisfaction on self-esteem

Life Satisfaction

Interpersonal relationship

(H14) Positive effect of interpersonal relationship on life satisfaction

Point 4: Conclusion: solid

Response 4: Thank you.

Reviewer 3 Report

This study investigates the predictive factors of academic achievement in a sample of students, using path analyses. Although the dataset may be interesting, I have some major concerns.

  • First of all, the writing could really be improved. Not only it is naïve, but there also are several typos and grammar errors. Although this is unrelated to the scientific quality of the manuscript, it catalyzes attention in a negative way.
  • The authors speak of an “interpersonal” factor but never provide a definition. Even in the methods section, they do not explain how the interpersonal score was derived and what it is expressing. A clear operational definition is needed (e.g., quantity of interpersonal relationships? Or their quality? Etc.).
  • Hypotheses are not clear: The term “included” is used several times, probably to express mediation, but it is not the most appropriate term. More importantly, the model hypothesizes very specific pathways among the variables, but no clear rationale is provided for these paths. The authors state that these hypothesized relationships depend on the fact that “previous evidence illustrated the inter-correlation among those factors”: this is not enough to justify the definition of such a complex model (rationale for almost any statistical model could be built upon previously observed correlations). In such a model with multiple mediators, one needs to indicate why variables are hypothesized to be in that specific position (e.g, why they are exogenous or endogenous). As presented now, it seems that the authors just “tried” this model by chance and found good indices of fit. This is my major concern: with cross-sectional data and in the absence of strong theoretical underpinnings for a model of path analysis, results run the risk of being totally uninterpretable and/or uninformative.   
  • I believe the statistical analysis section should speak of Structure Equation Modeling: this seems to be the technique employed by the authors. The estimator used in the model should also be included.
  • “The results from path analysis showed that the fit indices of all models were…”. How many SEM models were tested?
  • “…while depression (β = -0.119) had a negative direct effect in achievement”. This effect does not seem to be significant
  • Because of the absence of a clear rationale, the model is full of direct and indirect effects. For me, these results are puzzling and extremely hard to follow. Just to mention one example, consider the significant indirect effect of LIF→SEL→ANX→ACH. Why would this path make any clinical sense? Why would life-satisfaction “cause” self-esteem, which “causes” anxiety, which “causes” reduced achievement? For me, this is just a statistical artifact that depends on the correlations between the variables, but it’s hard to conclude something that informs or is informed by theory and clinical practice. Things are even more complex if this effect is considered in light of all additional direct and indirect paths in the model. In my opinion, a simple multiple regression model predicting achievement and evaluating the unique contribution of each predictive factor could be much more interpretable and easier to follow, in this case.
  • Limitations should also mention reverse causality. For example, the relationship between self-esteem and academic achievement may be reciprocal (students with higher academic achievement may have more positive self-views). The cross-sectional design of this study does not allow an investigation of reverse causality.  

Author Response

Response to Reviewer 3 Comments

This study investigates the predictive factors of academic achievement in a sample of students, using path analyses. Although the dataset may be interesting, I have some major concerns.

Point 1: First of all, the writing could really be improved. Not only it is naïve, but there also are several typos and grammar errors. Although this is unrelated to the scientific quality of the manuscript, it catalyzes attention in a negative way.

Response 1: We would like to apologize for the several typos and grammar errors. The revised manuscript was checked for grammar errors using Grammarly and then proofread and edited by the authors before resubmission.

Point 2: The authors speak of an “interpersonal” factor but never provide a definition. Even in the methods section, they do not explain how the interpersonal score was derived and what it is expressing. A clear operational definition is needed (e.g., quantity of interpersonal relationships? Or their quality? Etc.).

Response2 : Thank you for your suggestion. We added the definition of interpersonal relationship and the measurement procedure in the Methods section [Page 4, Lines 180 – 184] as follows.

“2.3.5. Interpersonal relationship

The interpersonal relationship, measured by an average point of 6 items, evaluated the respondent's relationship with other people (i.e., father, mothers, caregivers, siblings, teachers, and colleagues). Each item ranged from 0 (poor relationship) to 10 (excellent relationship). The reliability of this scale was 0.623.”

Point 3: Hypotheses are not clear: The term “included” is used several times, probably to express mediation, but it is not the most appropriate term. More importantly, the model hypothesizes very specific pathways among the variables, but no clear rationale is provided for these paths. The authors state that these hypothesized relationships depend on the fact that “previous evidence illustrated the inter-correlation among those factors”: this is not enough to justify the definition of such a complex model (rationale for almost any statistical model could be built upon previously observed correlations). In such a model with multiple mediators, one needs to indicate why variables are hypothesized to be in that specific position (e.g, why they are exogenous or endogenous). As presented now, it seems that the authors just “tried” this model by chance and found good indices of fit. This is my major concern: with cross-sectional data and in the absence of strong theoretical underpinnings for a model of path analysis, results run the risk of being totally uninterpretable and/or uninformative.  

Response 3: We would like to apologize for the vague statement and rationale of the hypothesized model. We considered both the hypotheses from previous evidence and our results according to correlation analysis. Since depression, anxiety, self-esteem, life satisfaction, and interpersonal relationship were reported as the factors influencing academic achievement; meanwhile, those factors also had inter-correlation with each other. Therefore, we suspected that the overlapping association might contribute as mediators to the indirect relationship to academic achievement in addition to their direct effect pathways. We have revised the rationale and objective of our study in the Introduction section [Page 2, Lines 52 – 59] as follows.

“The overlapping associations of self-esteem, life satisfaction, and interpersonal relationship to depression and anxiety may be mediators contributing to their influences on academic achievement. Previous studies which focused on finding factors that directly influence academic achievement may lead to the neglect of some indirect associations. Therefore, the aim of this study was to illustrate the association between study achievement and the potential influence of psychological factors among high school students using path analysis which allowed the investigation of both direct and indirect effects of each factor.”

Point 4: I believe the statistical analysis section should speak of Structure Equation Modeling: this seems to be the technique employed by the authors. The estimator used in the model should also be included.

Response 4: Thank you for your suggestion. The Path Analysis for our model was performed using the maximum likelihood estimator. Accordingly, we have added the related estimator in the Materials and Methods section [Page 5, Lines 203 – 208].

“Finally, the authors examined the validity of the model of academic achievement and five psychological factors, including depression (DEP), anxiety (ANX), self-esteem (SEL), life satisfaction (LIF), and interpersonal relationship (INT). The missing values of these variables were replaced with their means. All variables were included in the analysis to determine all plausible direct and indirect effect paths using Path Analysis (PA) with a maximum likelihood estimator.”

Point 5: “The results from path analysis showed that the fit indices of all models were…”. How many SEM models were tested?

Response 5: We would like to apologize for the missing explanation about the fit indices assessment. In particle, we only assessed the fit of a hypothesized path model. We have corrected that sentence accordingly [Page 7, Lines 245 – 247].

              “The results from path analysis showed that all fit indices of the hypothetical model were harmonious with the empirical data (Chi-square (c2) = 0.295 (df = 1), relative Chi-square (c2/df) = 0.295, RMSEA = 0.000, CFI = 1.000, TLI = 1.000, SRMR = 0.005).”

Point 6: “…while depression (β = -0.119) had a negative direct effect in achievement”. This effect does not seem to be significant

Response 6: We would like to apologize for the mistakes according to the results and interpretation. We have rechecked our dataset and found that some values in the scale variables, including depression, anxiety, and self-esteem, were invalid. Thus, we have reanalyzed and found that depression had a significant negative direct effect on academic achievement. The direct pathway of self-esteem on academic achievement was also changed to be non-significant. Meanwhile, other factors play a similar role to the former results. We have corrected the results presented in related figure and tables (Table 3 and Table 4).

Figure 1. The standardized regression coefficient of factors influencing Achievement and depression; * p<0.05, ** p<0.01,  R2 : Coefficient of Determination.

Table 3. Variables influencing depression and learning achievement of high school students.

ACH

DEP

ANX

SEL

LIF

INT

Achievement: ACH

1.000

Depression: DEP

-0.102

1.000

Anxiety: ANX

0.204**

0.522**

1.000

Self-Esteem: SEF

0.105

-0.535**

-0.407**

1.000

Life Satisfy: LIF

0.044

-0.421**

-0.279**

0.492**

1.000

Interpersonal relationship: INT

0.104

-0.317**

-0.139*

0.357**

0.466**

1.000

Mean

3.446

7.601

24.513

28.629

6.783

7.881

SD

0.460

4.276

13.331

4.285

2.067

1.687

Skewness

-0.825

0.615

0.542

-0.024

-0.505

0.468

Kurtosis

0.272

0.535

0.334

0.091

0.068

3.805

Tolerance

0.911

0.572

0.699

0.601

0.641

0.749

VIF

1.098

1.748

1.431

1.665

1.560

1.335

*p<0.05, ** p<0.001.

Table 4. Regression coefficients of factors associated with achievement, depression, and anxiety.

Path Directions

Direct  Effect

Indirect Effect

Total Effect

b

β

b

β

b

β

INTACH

0.011

0.042

0.017*

0.062*

0.028

0.104

                  INT→DEP→ACH

-

-

0.006

0.022

-

-

                  INT→ANX→ACH

-

-

0.005

0.018

-

-

                  INT→SEL→ACH

-

-

0.006

0.021

-

-

                  INT→ANX→DEP→ACH

-

-

-0.001

-0.004

-

-

                  INT→LIF→DEP→ACH

-

-

0.004

0.013

-

-

                  INT → SEL→DEP→ACH

-

-

0.003

0.010

-

-

                  INT → LIF→ANX→ACH

-

-

-0.006

-0.021

-

-

                  INT→SEL→ANX→ACH

-

-

-0.006*

-0.022*

-

-

                  INT →LIF→SEL→ACH

-

-

0.007

0.025

-

-

                  INT→LIF→ ANX →DEP→ACH

-

-

0.001

0.004

-

-

                  INT→ SEL→ ANX→DEP→ACH

-

-

0.001

0.005

-

-

                  INT→LIF→ SEL →DEP→ACH

-

-

0.003*

0.012*

-

-

                  INT→LIF→ SEL →ANX→ACH

-

-

-0.007**

-0.026**

-

-

                  INT→LIF→SEL→ANX→DEP→ACH

-

-

0.001*

0.005*

-

-

LIFACH

-

-

0.006

0.025

0.006

0.025

                  LIF → DEP→ACH

-

-

0.006

0.028

-

-

                  LIF →ANX→ACH

-

-

-0.010

-0.046

-

-

                  LIF→SEL→ACH

-

-

0.012

0.053

-

-

                  LIF→ANX→ DEP→ACH

-

-

0.002

0.009

-

-

                  LIF→SEL→ DEP→ACH

-

-

0.006*

0.026*

-

-

                  LIF→SEL→ANX→ACH

-

-

-0.013**

-0.057**

-

-

                  LIF→SEL→ANX → DEP→ACH

-

-

0.003*

0.012*

-

-

SELACH

0.014

0.127

-0.005

-0.046

0.009

0.081

                  SEL→DEP→ACH

-

-

0.007*

0.063**

-

-

                  SEL→ANX→ACH

-

-

-0.015**

-0.136**

-

-

                  SEL→ ANX→ DEP→ACH

-

-

0.003*

0.028*

ANX ACH

0.013**

0.375**

-0.003**

-0.076**

0.010**

0.299**

                  ANX → DEP→ACH

-

-

-0.003**

-0.076**

-

-

DEPACH

-0.023**

-0.216**

-

-

-0.023**

-0.216**

INTDEP

-0.264*

-0.104*

-0.541**

-0.214**

-0.805**

-0.317**

                  INT→ANX→DEP

-

-

0.043

0.017

-

-

                  INT→LIF→DEP

-

-

-0.154*

-0.061*

-

-

                  INT →SEL→DEP

-

-

-0.120*

-0.047*

-

-

                  INT→ LIF→ANX→DEP

-

-

-0.051

-0.020

-

-

                  INT→SEL→ANX→DEP

-

-

-0.053*

-0.021*

-

-

                  INT→ LIF→SEL→DEP

-

-

-0.143**

-0.056**

-

-

                  INT →LIF→SEL→ANX→DEP

-

-

-0.063**

-0.025**

-

-

LIFDEP

-0.270*

-0.131*

-0.450**

-0.218**

-0.720**

-0.348**

                  LIF→ANX→DEP

-

-

-0.090

-0.043

-

-

                  LIF→ SEL→DEP

-

-

-0.250**

-0.121**

-

-

                  LIF→ SEL→ANX→DEP

-

-

-0.111**

-0.054**

-

-

SELDEP

-0.290**

-0.290**

-0.128**

-0.129*

-0.418**

-0.419**

                  SEL→ANX→DEP

-

-

-0.128**

-0.129**

-

-

ANXDEP

0.113**

0.353**

-

-

0.113*

0.353*

INTANX

0.378

0.048

-1.478**

-0.187**

-1.100*

-0.139*

                  INT→LIF→ANX

-

-

-0.451

-0.057

-

-

                  INT →SEL→ANX

-

-

-0.468*

-0.059*

-

-

                  INT→ LIF→SEL→ANX

-

-

-0.558**

-0.071**

-

-

LIFANX

-0.791

-0.123

-0.977**

-0.151**

-1.768**

-0.274**

                  LIF→ SEL→ANX

-

-

-0.977**

-0.151**

-

-

SELANX

-1.132**

-0.364**

-

-

-1.132**

-0.364**

** p<0.01, * p<0.05. ACH, Achievement; ANX, Anxiety; DEP, Depression; INT, Interpersonal relationship; LIF, Life satisfaction; SEL, Self-Esteem.

We also revised the related interpretation and discussion in the Results and Discussion section accordingly.

Point 7: Because of the absence of a clear rationale, the model is full of direct and indirect effects. For me, these results are puzzling and extremely hard to follow. Just to mention one example, consider the significant indirect effect of LIF→SEL→ANX→ACH. Why would this path make any clinical sense? Why would life-satisfaction “cause” self-esteem, which “causes” anxiety, which “causes” reduced achievement? For me, this is just a statistical artifact that depends on the correlations between the variables, but it’s hard to conclude something that informs or is informed by theory and clinical practice. Things are even more complex if this effect is considered in light of all additional direct and indirect paths in the model. In my opinion, a simple multiple regression model predicting achievement and evaluating the unique contribution of each predictive factor could be much more interpretable and easier to follow, in this case.

Response 7: Thank you for your suggestion. We have revised the literature review, rationale, and objective of the study to be more apparent in the Introduction section, as mentioned in response to comment 9. In addition to the direct pathways between the studied factors, we are also interested in exploring the indirect effects resulting from all pathways according to our hypothetical model. Therefore, all plausible indirect pathways were included in the analysis and presented in Table 4. Additionally, we hypothesized that some studied factors might play the role as both independent and dependent variables. Therefore, we decide to use the Path Analysis model, which may relate to our objective than the multiple regression model. We have addressed the use of Path Analysis in our study in the Materials and Methods section accordingly [Page 5, Lines 206 – 208].

“All variables were included in the analysis to determine all plausible direct and indirect effect paths using Path Analysis (PA) with a maximum likelihood estimator.”

Point 8: Limitations should also mention reverse causality. For example, the relationship between self-esteem and academic achievement may be reciprocal (students with higher academic achievement may have more positive self-views). The cross-sectional design of this study does not allow an investigation of reverse causality.

Response 8: Thank you for your suggestion. We have added this limitation to the Discussion section accordingly [Page 12, Lines 380 – 384].

              “In addition, some factors might have the reverse causality, such as the reciprocal relationship between self-esteem and academic achievement (students with higher academic achievement may influence the positive self-views). However, we did not investigate the reverse causality in our analyses due to the limitation of the cross-sectional design, which does not allow an investigation of reverse causality.”

Round 2

Reviewer 2 Report

Dear authors,

Thank you for the revised document.

Author Response

Response to Reviewer 2 Comments

Point 1: Thank you for the revised document.

Response 1: Thank you for your previous comments which is useful for the revision.